# Geo-Enabled Sustainable Municipal Energy Planning for Comprehensive Accessibility: A Case in the New Federal Context of Nepal

**Hari Krishna Dhonju** [1,*] , **Bikash Uprety** [2] **and Wen Xiao** [3]

1   Pathway Technologies and Services Pvt. Ltd., Kathmandu 44600, Nepal
2   REEEP, GIZ, Kathmandu 65760, Nepal; bikash.uprety@giz.de
3   School of Geography and Information Engineering, China University of Geosciences, Wuhan 430074, China; wen.xiao@cug.edu.cn
*   Correspondence: dhonjuh@pathway.com.np; Tel.: +977-9841402416

**Abstract:** Energy is a fundamental need of modern society and a basis for economic and social development, and one of the major Sustainable Development Goals (SDG), particularly SDG7. However, the UN's SDG Report 2021 betrays millions of people living without electricity and one-third of the world's population deprived of using modern energy cooking services (MECS) through access to electricity. Achieving the SDG7 requires standard approaches and tools that effectively address the geographical, infrastructural, and socioeconomic characteristics of a (rural) municipality of Nepal. Furthermore, Nepal's Constitution 2015 incorporated a federal system under the purview of a municipality as the local government that has been given the mandate to ensure electricity access and clean energy. To address this, a methodology is developed for local government planning in Nepal in order to identify the optimal mix of electrification options by conducting a detailed geospatial analysis of renewable energy (RE) technologies by exploring accessibility and availability ranging from grid extensions to mini-grid and off-grid solutions, based on (a) life cycle cost and (b) levelized cost of energy. During energy assessment, geospatial and socio-economic data are coupled with household and community level data collected from a mobile survey app, and are exploited to garner energy status-quo and enable local governments to assess the existing situation of energy access/availability and planning. In summary, this paper presents a geo-enabled municipal energy planning method and a comprehensive toolkit to facilitate sustainable energy access to local people.

**Keywords:** geospatial; grid sampling; renewable energy; energy accessibility; best available technology; municipal energy planning

## 1. Introduction

Nowadays, the use of energy varies from cooking, lighting, heating, communication, and operating home appliances to running institutions, enterprises, and industries that can improve the quality of life [1]. Energy is crucial to performing the daily socio-economic activities of modern society and its consumption is one of the key indexes of human development, as per capita energy consumption of developed countries outweigh that of under developed countries like Nepal [2]. The United Nations introduced Sustainable Development Goals (SDG) and emphasized SDG7 to ensure sustainable, affordable, and modern energy for all [3]. Although the electricity access rate has increased from 83% in 2010 to 90% in 2019, millions of people still live with no access to electricity, and one-third of the world's population is without clean cooking technologies and systems [4]. Moreover, the ongoing pandemic has slowed down the global progress in universal access to energy and caused millions of people who recently gained access to electricity to lose the ability to pay.

The Constitution of Nepal 2015 has incorporated a federal structure comprising three tiers of government that devolves power from central to the local governments of 753 municipalities and 7 provinces. The newly formed local governments are now mandated to design policies, support programs, and deliver public goods and services regarding decentralized renewable energy (DRE) [5]. Due to the rapid urbanization and merging of municipalities that form 293 urban and 460 rural municipalities, only 37.8% of the population reside in remote areas. In 2019, the Nepal Electricity Authority (NEA) and Alternative Energy Promotion Center (AEPC) claimed that Nepal has a total of 87.55% households with electricity access, with 77.80% from the grid and 9.75% from DRE, and 12.45% without any access [6]. The disconnected households corresponding to villages or settlements in rural areas are sparsely distributed or isolated in a small household cluster. Poor accessibility to these locations, challenging terrain, and topographic diversity could be some of the reasons behind the hindrance to penetration of grid electricity in such areas. Furthermore, 69% of households still rely on solid biomass for cooking and a majority of the households (68.6%) use traditional fuels as their primary stove for cooking and heating [6]. Alongside the adverse effects on health and gender, the climate is adversely impacted by traditional cooking practices. This also means that Nepal remains far off track from achieving the 2030 target for SDG, 7.1: ensuring universal access to affordable, reliable, and modern energy cooking services (MECS). To address this, the possible best available renewable energy (RE) has to be assessed and the municipalities are to be strengthened and empowered through evidence-based planning, informed decision making, and better allocation of local resources. Moreover, both grid and off-grid solutions are vital for achieving universal access and have cross-sectional benefits on a wide range of other SDG indicators, such as health, education, gender equality, livelihoods, and poverty reduction. In order to realize the benefits, and coordination between all three tiers of the government in the federal structure of Nepal, it is essential to adopt an integrated planning approach to achieve universal access to energy for the citizens of Nepal.

Renewable energy, as a clean form of energy, comes from natural sources or processes that are constantly replenished. The primary RE sources are hydropower, solar, wind, and biomass, which are abundantly available in Nepal. Harnessing a mix of technologies is the critical determinant to provide energy access in the far-flung communities of Nepal. KC et al. [7] present the current status of RE, which helps to identify major RE sources for the promotion and adoption of RE technologies (RETs). Shrestha [8] claimed that Nepal has about 42,000 MW worth of economically exploitable hydropower potential, out of which only 2% has been installed, and that micro-hydropower (<100 kW), whose potential installation capacity has been estimated to be 50 MW, can be installed in the vicinity of the local environments.

For more than two decades, the government's nodal agency, the AEPC, together with various development partners, has been scaling up DRE with subsidies to stimulate demand for off-grid RE in the country [9], which lacks an appropriate energy assessment tool. This fiscal tool coupled with community mobilization practices helped the uptake of various RETs that have transformed the lives of millions of poor households by providing them cooking, lighting, and other energy-induced income-generating solutions. Specifically, micro/mini hydropower plants are contributing to uplifting the livelihoods of rural people and opening up other avenues for economic activities. With the support of the AEPC, more than 32.159 MW of micro/mini hydropower plants have been providing off-grid electricity to more than 3.5% of the population of Nepal [6]. Additionally, Nepal has a high potential for solar energy. Nepal experiences an average of 6.8 sunshine hours per day with an average intensity of solar energy of 4.7 $kW/m^2/day$ (ranging from 3.9 to 5.1 $kWh/m^2/day$) [10,11]. Moreover, reaching universal electricity access to all is still a significant challenge as the grid networks are expensive for a hilly country like Nepal due to its undulated topography and scattered rural settlements with a lack of accessible roads and adequate transmission networks [7]. Further, for locations where grid extension is not feasible, DRE such as solar mini-grids, a solar home system, or micro-hydropower

can be technologically and economically feasible and sustainable. Hence, it is paramount to have a comprehensive local energy plan with the exploitation of decentralized RETs to electrify rural areas based on possible energy sources and to improve the quality of life of the population residing in far-flung communities. Additionally, it enables local municipalities to understand the energy accessibility, monitor development, and make informed policy decisions and helps them to plan and promote the best suitable RETs to meet the local needs and demands.

Nowadays, several free geospatial datasets, such as administrative municipal boundaries, the Shuttle Radar Topography Mission (SRTM), and landcover are becoming readily available in the public domains [12–14]. At the same time, recent development in free and open-source software (FOSS)-based geospatial information and communication technology (GeoICT) offers a historic opportunity to enable a fast-changing and knowledge-driven society. GeoICT and tools are capable of collating, analyzing, visualizing, and even online processing complex algorithms compared to traditional data management and processing systems [15–17]. The application of these tools and technology is promising in assessing energy solutions among various RE alternatives as demonstrated by the capabilities of GeoICT [18–22] in various disciplines.

The main aim of the paper is to examine municipal planning processes to facilitate sustainable and comprehensive energy access to the citizens in the jurisdiction of local governments. This aim is split into two sub-objectives: (a) to present the municipal energy planning (MEP) process implemented by the local government, and (b) to propose a geo-enabled MEP toolkit based on the objective to assure more people have access to modern and high-quality energy access in isolated communities. The planning process is discussed in the multi-tier framework (MTF) [23] for energy access measurements. The main purpose of the MEP toolkit is to facilitate the generation of a comprehensive local energy planning report in a data-driven and evidence-based approach by utilizing geospatial datasets, tools, and technologies. The MEP toolkit devises primarily three base tools: (a) spatial grid sampling and energy baseline survey; (b) best available technology (BAT); and (c) a tool to check micro-hydropower plant potential (MHPP) to generate the comprehensive municipal energy plan.

## 2. Literature Review

In recent days, energy planning has been accomplished using energy models. An energy model can be defined as a simplified representation of a real system that can be used to conduct complex analyses and/or calculations. Hourcade et al. [24] and Grubb et al. [25] classified different energy models in terms of purpose, structure, optimization, aggregation, geographical coverage, and so forth. Energy optimization models which are used to identify the best among the available alternatives for promoting the use of RE sources are presented by [26,27]. These studies demonstrated the use of energy plans developed based on supply and demand at cluster to municipal levels.

Generally, energy planning is carried out at a centralized level because of grid electricity and its distribution which triggers remote villages to remain isolated due to the high incurring the capital cost of generation and transmission. However, it fails to solve the problems of rural areas where the population is scattered, and village settlements are isolated. To address these sorts of bottlenecks, Das et al. [28] studied selecting an appropriate alternative energy technology with a priority which can harness RE sources for rural areas of India. With the use of dynamic programming, the study was carried out based on a field survey. Similarly, Van Beeck et al. [29] proposed a method for local energy planning, at village level, which focuses on preparing an energy plan based on data availability and RE options. Although the method uses a decision support tool, it does not help energy planners to take recommended actions.

On the other hand, energy inputs are essential parameters for comprehensive analysis of energy scenarios of a rural system which shall depend on primary and secondary data sources [30]. Because of the current pattern of centralized electricity planning, Hire-

math et al. [31] reviewed different decentralized energy planning models and approaches concerning locally available energy sources. The review emphasizes local level and decentralized energy planning models to adopt a bottom-up approach using fragmented and isolated data to solve the problems of rural areas regarding access to energy. Emphasis is given to the fact that large proportions of the rural population depend on low-quality energy sources leading to a low quality of life which leads to environmental degradation. Thus, the review highlights a need for an alternate approach or tool for RE planning for sustainable economic development.

In the context of Nepal, a majority of the Nepalese population resides in rural areas using traditional energy sources for lighting and cooking. These energy sources are not clean, sustainable, or techno-economically feasible, and the use of which adversely affects the health of household members, particularly women and children [32] resulting in limited active working hours per day [33] for performing better economic activities.

Where national grid energy is inaccessible or costly when extending it to rural areas, DRE can be a significant source that can be exploited for a single targeted house (solar home system) or a group of houses (mini-grid). Moreover, Nepal is blessed with snow-fed and natural river systems with the potential for micro-hydro power installations locally. Typically, energy planning is explicitly spatial as stated in Herrmann et al. [34], who presented decentralized planning concepts for examining at a regional and local level (a settlement as a community) using a geographic information system (GIS) and modelling approach. On the other hand, geospatial data related to energy and topography [13,14] are readily available and the data gaps pertaining to ground reality can be acquired with the use of the latest geospatial tools and technologies [35]. Kusiak [36] claimed that new ideas can be generated by evaluating such datasets, e.g., transforming data into knowledge [37], through a data-driven approach to innovation by creating a specific service. In other words, these datasets can be utilized to conduct an energy assessment to produce a comprehensive energy plan from the available alternative RE sources at a local level. At the same time, a comprehensive MEP report is generated for medium term energy planning.

Though energy planning is required at all scales, it is generally conducted at either national or regional level given in view of supply and demand, thus not ensuring sustainability and accessibility. Mentis et al. [38] discussed a complementary GIS-based approach to already existing energy planning models. However, there is no evidence of a generic energy assessment and planning tool which can assess energy and generate energy plans at the local or community level. Further, Chicco [37] argues data-driven analysis is conducted based on field data which provides ground truth information. Municipalities are facing challenges in the transition to RE systems and hence data is collected across the case municipalities towards the development of energy assessment under the design tool principal [39].

Numerous studies [40–44] developed a set of electrification planning tools and techniques, primarily focused on extending the national grid for electricity access, and later compared to decentralized alternatives. Balderrama et al. [41] proposed a geospatial electrification methodology, also known as the open source spatial electrification tool (OnSSET), for large scale electrification modelling by identifying the priority areas for micro-grids with load simulation and optimization techniques for rural electrification planning. Korkovelos et al. [45] in Malawi and Mentis et al. [46] conducted case studies for electrical planning in sub-Saharan Africa based on the OnSSET, focusing on the role of open access data to achieve SDG7. Ciller et al. [40] discussed a computer-based optimization model for automatic electrification planning to identify the lowest cost electrification system in rural areas, also known as the reference electrification model (REM) [47], which covers analysis for grid extension, off-grid mini-grid, or standalone systems of any given population size. Kemausuor et al. [42] used the network planner for electricity planning in Ghana, which can predict the costs of different electricity generation technologies of the un-electrified communities and gives freedom to choose the most cost-effective technology in the given conditions. In addition, Korkovelos et al. [48] explored a least-cost electrification model

in Afghanistan and found cost-optimal electrification mix was very sensitive to a local context, e.g., security and power supply, and varies from location to location [43]. Further, Mentis et al. [49] discussed needs and constraints, e.g., access to data and analytical tools, besides identifying technologies and investment needs for new customers and introduced data and methods in an online and interactive digital platform, named Energy Access Explorer. However, these existing tools and methodology lack a comprehensive energy plan which can be implemented at local community level.

In summary, a geo-enabled energy assessment tool is still lacking at a local level, even in a small village community with proper data collation from various sources. In this paper, energy assessment is performed at the settlement or community level with the use of available energy and topographic geospatial datasets and complementary in-situ data collection using the latest FOSS-based geospatial tools and technologies. The energy assessments at geographically isolated communities are aggregated at the municipal level. This study presents a novel method to generate a comprehensive municipal energy plan that automatically produces maps and info graphs.

## 3. Materials and Methods

### 3.1. Data

#### 3.1.1. Data Source

Various geospatial datasets have been used to check the potentiality of each available RE option: (a) grid extension, (b) mini-grid (MG), and (c) solar home system (SHS). For example, municipal administrative boundary [50] has been used to limit energy assessment; building footprints from OpenStreetMap (OSM) [51] were used to discriminate contagious households as an isolated settlement or a household cluster; SRTM data [13] is used to derive river networks and topographic variables like elevation, slope, and aspect including computation of solar energy potential; use of grid electricity distribution lines (33 kV and 11 kV) [52] and road networks [53] for checking access to the national grid; population and demographic [54,55] for estimating energy demand; landcover data [56] for differentiating settlement from the forest, agriculture, and other land cover types. These datasets are becoming freely available locally [50,53,54] and globally [13,51,57]. From these datasets, energy assessment ready datasets were further derived, for example, household clusters, river networks, and energy potential datasets. The derivation of these datasets is further discussed in the following sections.

#### 3.1.2. Deriving Energy Datasets

**Household clusters:** Generally, according to the size of the individual building footprint obtained from OpenStreetMap (OSM) [51], a household ranges from 3 to 4 m in hilly terrains of rural Nepal. If two building footprints are separated within 100 m of aerial distance, justifiable in hilly geography, both footprints are treated within the same cluster. Thus, formed household clusters will help to visualize and understand how many people are living with or without access to energy, for example, from the national grid.

**River networks:** River networks of the whole of Nepal were generated from 30 m of SRTM data [13]. The dataset was topologically corrected, and river order was calculated for each of the river lines with elevation nodes assigned. Thus, the created dataset was used to explore whether micro-hydro power can be installed for a given river stream at the municipality level.

**Grid extension potential:** Grid electricity distribution lines (33 kV and 11 kV) were topologically corrected. Topographic cost distance was computed for these electricity distribution lines. A 1000 m cut-off cost distance was used to define the potential grid extension area which helps to check whether grid electricity is extensible or not to a given settlement cluster. The decision to extend a grid is made if a given point within this region is marked as grid extensible, otherwise it is not.

**Micro-hydropower potential:** A 1000 m aerial radial distance was used as a catchment area for existing micro-hydropower. If a point lies within this area, it is marked as accessible to micro-hydropower, otherwise it is marked as not accessible.

**Solar potential:** A 90 m SRTM dataset [13] was used for potential solar mapping. Solar radiation was calculated at pixel level and flagged as solar potential if it met the criteria of at least 4 sunshine hours and annual sunshine days greater than 300. This dataset helps to check whether there is solar potential for a given point or not for installing SHS or solar MG.

### 3.1.3. Clean Cooking Framework

In line with the energy planning process, REEEP developed a simplified clean cooking access measurement framework towards clean cooking solutions which defines criteria to categorize fuels and stoves. The framework classifies cooking fuel into three categories: (a) basic fuels (e.g., coal and kerosene), (b) intermediate fuels (e.g., biomass products), and (c) modern fuels (e.g., BLEN: Biogas, LPG, electricity, and natural gas). Similarly, it categorizes household cookstoves into basic, intermediate, and modern cookstoves. Further, access levels are defined as (a) tier 0: low, (b) tier 1–2: partial, (c) tier 2–3: intermediate, and (d) tier 4: modern and advanced following the multi-tier framework (MTF) [23]. This has been implemented to conduct a field-based survey as well as to establish the energy baseline of the given municipality.

### 3.2. Methods

### 3.2.1. Data Aggregation

Various fragmented and segregated data were compiled and integrated from different sources into a single gateway. It includes (a) population and demographic [54], (b) topographic [13,50,53,56], (c) energy infrastructure [52], (d) crowd sourced [51], and (e) field-based survey data. Population and demographic data were used to study the trend of population growth and enumeration of households. Topographic data comprised of administrative boundary, elevation, landcover, river and road networks, and settlements.

An administrative boundary was used to discriminate municipal boundaries and limit the energy assessment within a municipality. Elevation data was essential for understanding the topography of the location in terms of altitude, slope, and aspect of the terrain where energy assessment is to be conducted. Moreover, potential solar areas were derived using the elevation data. Landcover data helped to understand ground situations such as forest, agriculture, and built-up areas. Potential micro-hydropower was explored based on the availability of river networks within a municipality. Accessibility of municipality was assessed with the use of road networks, topography, and settlement locations providing information on where the people reside. Energy infrastructure was understood through considering locations of sub-stations, distribution lines, and locations of transformers, and this information was assessed to determine the accessibility of grid energy and the possibility of its extension. Crowd-sourced data such as building footprints or locations of households were most essential for delineating the settlement or to know where people live. Such data were used to estimate the catchment area of an energy plant (solar mini-grid or micro-hydropower) which could be installed locally. The data gaps from these secondary sources were complimented by field-based surveys, for example, determining energy baseline using mobile survey app. By integrating geospatial data with demographic and development scenarios, the developed toolkit provides fact-based visualizations and projections for local energy planning. By aggregating data to higher levels, this toolkit provides relevant inputs to provincial and federal governments for energy planning, monitoring, and policy making.

Thus, these datasets assisted in getting preliminary energy-related information for any municipality. Further, this enabled us to explore, visualize, and acquaint ourselves with the energy situation and available energy resources of a particular interest area, even at a

community level. Moreover, an energy assessment could be performed in a data-driven approach specific to the spatial location and viable contextual information.

### 3.2.2. Energy Assessment Methods

Most of the households in the country receive electricity from the national grid and about 9.75% are from decentralized renewable sources [6] such as pico-hydropower (<10 kW), micro-hydropower (10–100 kW), and SHS. Although there is vast potential for mini-hydropower (>100 kW and <1000 kW), very few projects have been developed in the country in recent years due to technical and operational challenges. The most appropriate technology depends on specific circumstances. Therefore, exploration and assessment of various energy alternatives are required to choose a BAT in terms of investment and energy cost.

To assess the cost-effectiveness of the technological options, two energy cost assessment methods are devised: (a) life-cycle cost (LCC) [58] and (b) Levelized Cost of Energy (LCOE) [59]. The LCC of a project is defined as the total amount of all costs incurred in the project from its initial design stages to its decommissioning. It is calculated as the present value of the total cost of purchasing, installing, operating, maintaining, and repairing energy generating and/or distributing system over its economic life and is given by Equation (1). The LCC analysis takes all costs into account, from the costs of construction, fuel costs, and repair costs, to the costs imposed by emissions from the project. The LCC analysis of a project helps to compare different technical options to determine which technology is the cheapest in the long run.

$$LCC = CC + \sum_{i=0}^{n} \frac{(OMC + RC + FC)}{(1 + DR)^i} \tag{1}$$

where:

    *LCC* = Life cycle cost
    *CC* = Capital cost
    *OMC* = Operation and maintenance cost
    *RC* = Replacement cost
    *FC* = Fuel cost
    *DR* = Discount rate factor
    *n* = Number of years

The *LCC* analysis is usually presented in the form of the levelized cost of energy (LCOE) of the project. *LCOE* is defined as a measure of a power source that allows the consistent comparison of different methods of electricity generation. It is an economic assessment of the average total cost to build and operate a power-generating asset over its lifetime divided by the total energy output of the asset and can be calculated using Equation (2).

$$LCOE = \frac{LCC_i}{\sum_{i=0}^{n} \frac{EG_i}{(1+DR)^i}} \tag{2}$$

where:

    *LCOE* = Levelized cost of energy
    *LCC* = Life cycle cost
    *EG* = Electricity generation
    *DR* = Discount rate factor
    *n* = Number of service years

The *LCOE* measures the costs over the lifetime of the proposed energy project and determines how much it costs to produce an amount of energy per kWh. The *LCC* analysis and *LCOE* of the project can be used to compare the lifetime costs of one energy option to another to identify the BAT.

In the context of the energy white paper [60] of Nepal, energy access should be with the provision of tier-3 energy access option in line with MTF and hence it is considered as

the baseline and needs to be adopted. The amount of energy within tier-3 will be adequate for various appliances such as electrical lighting, air circulation, television, and phone charging. The power and energy consumption will be a minimum of 200 W and 1.0 kWh, respectively, with a duration of 8 h per day minimum, and a minimum 3 h in the evening to meet this tier. To assess and identify the energy options, the three main RE options were considered, namely, (a) grid-extension, (b) mini-grid (micro-hydropower or solar mini-grid), and (c) solar home system (SHS). To check the potentiality and possibility of these energy options, the following criteria have been set.

**Grid-extension (GE):** Grid extension was evaluated with an estimation of LCC and LCOE based on 11 kV gridline, finding the nearest transformer from the catchment areas such as the centroid of a settlement cluster and aerial distance between them targeting tier-3 energy by default.

**Mini-grid (MG—micro-hydropower or solar mini-grid):** Firstly, for the given location, it was checked whether there is already an MG or provision for the near future for development; otherwise, assess the possibility for development of a potential mini-grid in the vicinity of the area or its surroundings within a municipality.

**Solar home system (SHS):** The calculation also provides LCC and LCOE for SHS for a single household. The solar potential of the area was determined with a criterion of minimum peak sunshine of 4 h assuming annual sunshine days of greater than 300 days.

### 3.3. Decision Tree

Just looking at the technological aspect of each energy option and associated costs shall not be sufficient to decide which energy solution should be chosen among the available energy options for a particular area of interest. For determining the BAT, a decision tree (Figure 1), developed by the AEPC, was modified with the introduction of levelized cost of energy (LCOE) and adopted. According to this decision tree, at first, energy demand (kWh/year) was estimated for a given location or a settlement cluster. Then, the possibility of grid extension from the nearest grid (distribution line or a transformer) was checked. If the grid extension limit is greater than the allowable distance from the grid, a plan for a future plan for grid extension is reviewed, and if the future grid extension period is less than five years, it is recommended to go for SHS or MG for the given grid extension grid. If not, energy assessment for each option was evaluated in terms of LCC and LCOE, and technology with the least LCC and LOCE was selected as the feasible BAT.

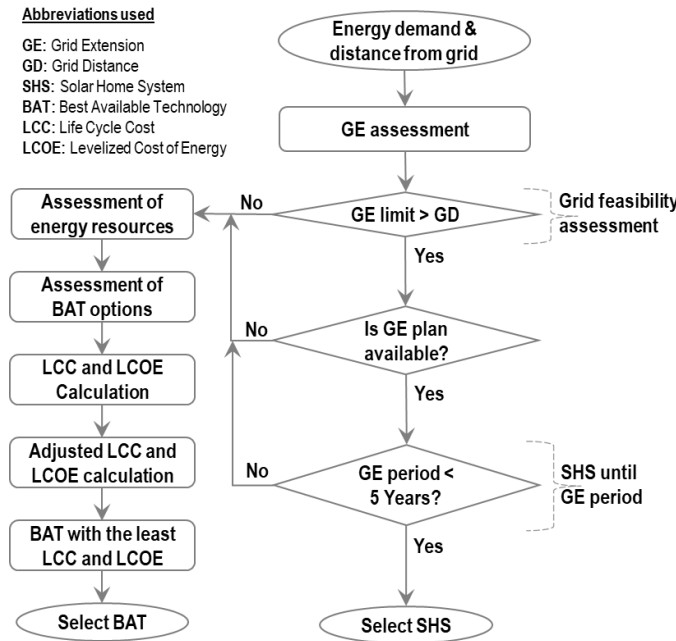

**Figure 1.** Modified decision tree from the AEPC.

## 4. Implementation

### 4.1. Energy Assessment Tools

The energy planning process consists of a set of energy assessment tools known as the Municipal Energy Planning (MEP) toolkit. The toolkit includes (a) a dashboard for exploring geospatial data and infographics; (b) a grid-sampling and survey tool for conducting a household, communities, enterprises, and energy infrastructure survey; (c) BAT for identifying the best technology among available renewable energy options; (d) micro-hydropower potential exploring tool; and (e) a report generation module (RGM) that compiles all information and generates a comprehensive periodic municipal energy plan.

**Dashboard:** A platform that facilitates the exploration of different geospatial datasets and other information related to population, energy and physical infrastructure, landcover, and field survey-based baseline energy information including clean cooking infographics. The tool can be used to explore and visualize data that supports evidence-based energy planning for the three tiers of government.

**Grid-sampling and survey tool:** While conducting a field-level household survey, visiting every household might not be practical as it requires many resources and can be time-consuming and costly. In this case, a sample-based survey can be handy and efficient, and the grid sample [61] tool has been devised to conduct field surveys for representative households only. From this method, grid samples are generated at the municipal level, ensuring that each ward has at least one grid sample. Thus, a household survey is carried out in each of these grid samples and later information is aggregated at the municipality level [62]. Survey forms were designed with the ODK platform [35] and were integrated with the MEP tool to conduct field surveys to capture data of (a) households, (b) focus group discussions at ward level, (c) energy infrastructure, and (d) enterprises.

**Best available technology (BAT):** Identifying the best energy technology among the available renewable energy options is termed BAT. The most appropriate technology is evaluated based on energy demand, availability of energy resources, minimal LCC and minimal LCOE, and access to energy for a specific geographic location. The LCC and LCOE is calculated by using Equations (1) and (2), respectively.

**Micro-hydropower potential (MHPP):** If the net head, discharge, turbine/generator efficiency is given, power generation can be estimated using the potential energy of water using Equation (3) [63]. Reversely, for the given geolocation of intake of a specified river, discharge and required power, downstream location can be estimated with a promising head availability and is calculated by using Equation (4) following the river creek line from intake to the location of the head.

$$P = \eta \rho g H Q \qquad (3)$$

where:

$P$ = the power output, measured in Watts.

$\eta$ = the efficiency of the turbine

$\rho$ = the density of water, taken as 1000 kg/m$^3$

$g$ = the acceleration of gravity, equal to 9.81 m/s$^2$

$H$ = the head, or the usable fall height expressed in meters

$Q$ = the discharge, also called the flow rate, calculated in m$^3$/s

With values of $\eta$ = 0.7 (70% efficiency) for instance, $\rho$ and $g$ used, the Equation can be generalized to the Equation (4).

$$H = \frac{P}{6867Q} \qquad (4)$$

Additionally, this tool can be used to calculate a circular catchment area that encompasses the number of underlying households which can be served from the generated energy.

**Report generation module:** All the information and results generated during the energy processes are compiled automatically into a printable report, a periodic municipal energy plan. The report consists of (a) characteristics of a planned municipality; (b) baseline

energy profile; (c) energy assessment results; (d) comprehensive maps; (e) an executable plan with prioritized activities; and (f) monitoring and evaluation framework.

Characteristics of the municipality contain information on demography, physical and energy infrastructures, and topography. Infographics including clean cooking generated from the field survey is presented in the baseline energy profile. BAT results are summarized in the energy assessment section of the report. A list of automatically generated illustrative maps are presented in relevant sections. Activities are prioritized based on ranking and availability of funding resources and presented in a plan. For the implementation of the generated plan, a monitoring and evaluation framework is appended as a part of the energy planning report.

*4.2. Development Framework*

Recently, the adoption and development of FOSS has been gaining momentum among governmental and non-governmental organizations. One of the primary reasons for choosing this technology is that it is freely licensed software to use, copy, study, and change or extend, and open-source code which will abide by black-box algorithms and help to build trust among the people and implementers. At the same time, the adoption of FOSS can reduce the cost of software development and operation, increase security and stability, and be free from possible vendor locks. On the other hand, this practice creates employment opportunities in software development. Therefore, free and open-source tools and technologies have been used to develop the software system. This will help to minimize the cost of ownership as well as up-gradation of the system which will make the system sustainable in terms of implementation and operation in the long run. Furthermore, it is necessary that the tools and technologies to be used should be mature and well-developed, and should follow industry standards. The technologies should bear high-security standards, be stable, and should have been adopted by the broader communities. Because of this, a FOSS-based development framework has been chosen. With this background, the framework used the following software development tools and geospatial technologies, which are grouped into three major tiers: (a) database, (b) backend, and (c) frontend.

(a) **Database:** The database tier can be considered a physical data layer which is the lowest layer of the whole system and is treated as a foundation layer for the rest of the tiers. This layer provides data services to a higher level and holds various thematic, supportive, and master databases, including harvesting of geospatial datasets guided by policy and standards [64]. The database has been implemented in Postgres/PostGIS [17].

(b) **Backend:** The main engine of the system was developed based on the backend development as the application layer. It consists of various web-based application programming interfaces (APIs) and server-side applications hosted in the servers are required for frontend development, integration, and interfacing. The APIs are used to serve data, map and geoprocessing services, enabling developers to work towards progressive application development. Django framework was used for the development of APIs for serving data and promoting geoprocessing services. Open Data Kit (ODK) platform [35] was used for mobile data collection at the household level and others. GeoServer was used for web mapping. R scripting tool was used for implementation of grid-sampling tool [61]. Geospatial Data Abstraction Library (GDAL) was used for geospatial data processing, and all were hosted in Linux environment with NGINX web server [15–17,65].

(c) **Frontend:** A presentation layer comprises various user interfaces where web users interact and play with web interfaces quickly and interactively in a user-friendly environment. This layer constitutes data, map and geoprocessing services through APIs from the backend layer. The tools and technologies used are OpenLayers for client-side mapping and Vue.js with HTML, JavaScript, and CSS for web page rendering.

### 4.3. Case Study: An Example in Nepal

A case study, taking an example of a municipality in Nepal, is presented for the process of MEP in the case of Khanikhola rural municipality, Kabhrepalanchowk district, Bagmati province, specifically by running the tools for grid sampling, best available technology, and micro-hydropower potential. Initially, settlement clusters were identified with no access or less access to the national grid. Then, these tools are executed repeatedly at different settlement clusters to cover the municipality. The municipal energy planning (MEP) toolkit can be accessed via http://mep.pathway.com.np/ (accessed on 21 March 2022). Dashboard and planning mode of the MEP toolkit are shown in Figures 2 and 3. Three significant tools: (a) grid sampling, (b) best available technology (BAT), and (c) micro-hydropower potential (MHPP) tools are discussed in the succeeding sections.

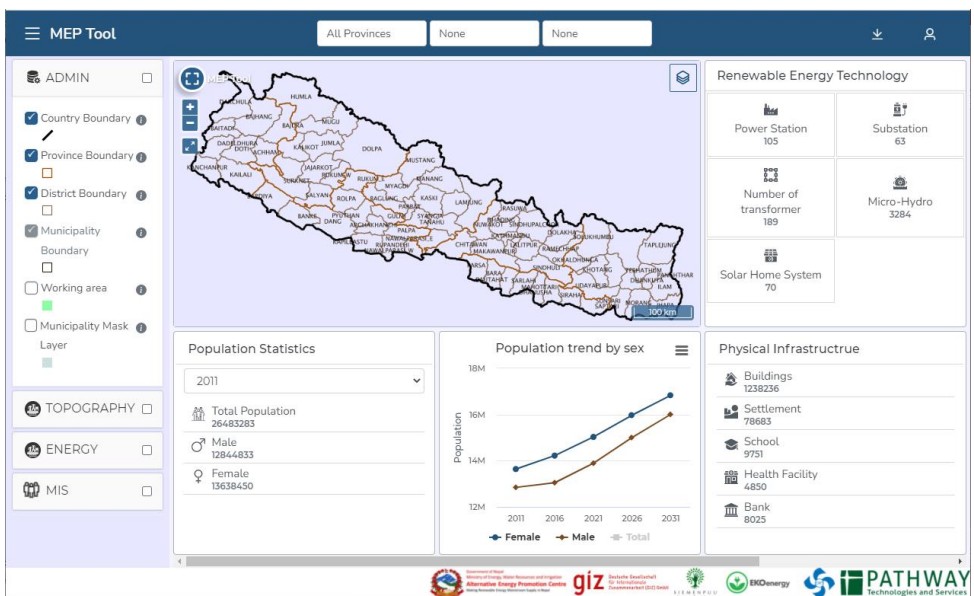

**Figure 2.** Dashboard mode of Municipal Energy Planning Toolkit.

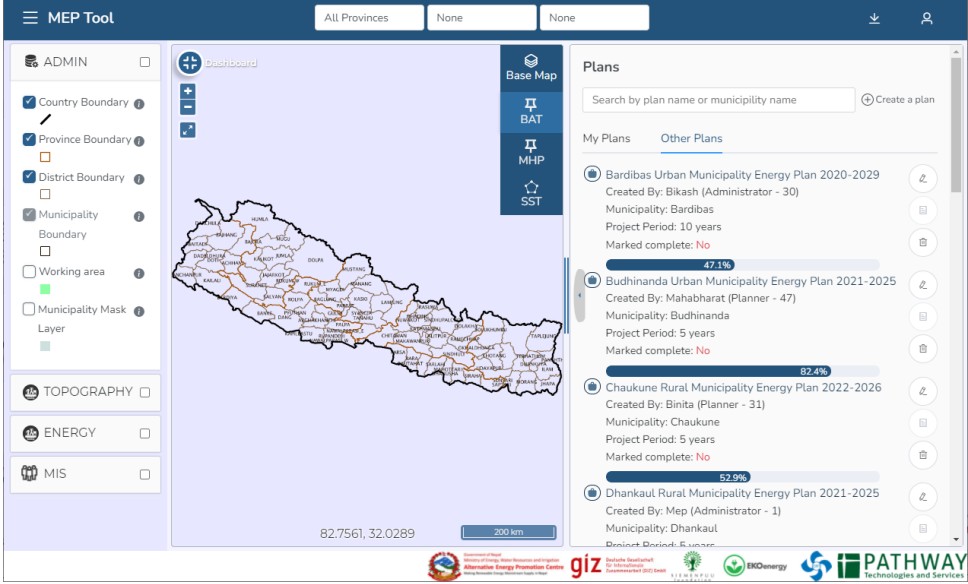

**Figure 3.** Planning mode of Municipal Energy Planning toolkit.

**Grid-sampling tool (GST):** For implementing the grid sample tool, a 100 m grided WorldPop [55], rural and urban areas (municipalities) based on the current federal structure of Nepal, and municipal boundaries have been used. The size of the grid sample used is 500 m, which constitutes 100 people and 10 households in each of 40 grid samples as the clusters, i.e., representing the whole municipality by 400 households, as suggested by Pelz [62]. A sample grid generated with these parameters for the municipality has been presented in Figure 4. Municipal boundary and squared grid samples are displayed in green and yellow colors, respectively.

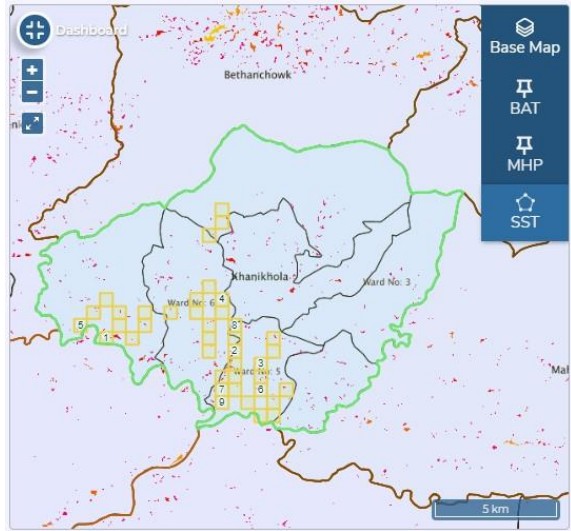

**Figure 4.** Distribution of grid samples (municipal boundary: green; ward boundary: black; and grid sample: yellow).

**Best available tool (BAT):** Settlement cluster is identified by exploring household cluster data and a polygon is digitized to encompass this cluster as shown in Figure 5, then BAT is executed for this cluster on the server-side. The settlement cluster is represented by a red household patch encompassed by a digitized cyan color boundary. With a successful run of the BAT, the system presents graphical scenarios for LCCs and LCOEs for all available RE options and recommends a BAT for tier-3 electricity access based on the inbuilt algorithm, the decision tree. For this example, the tool recommends SHS as the best feasible technology for electricity access for the given settlement cluster.

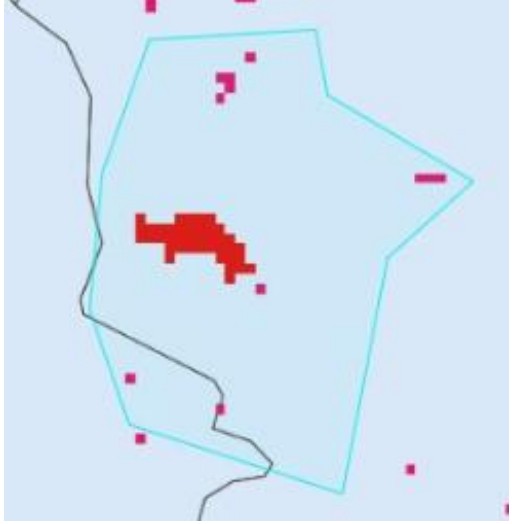

**Figure 5.** A settlement cluster (cluster boundary: cyan; a settlement cluster: red patch).

**Micro-hydropower potential:** Micro-hydropower potential was explored by clicking an intake point (blue) nearby a settlement of Mahadevtar municipality along Mul Khola (a creek line of a stream) with a parameter of discharge (0.02 m$^3$/s) and at an interval of 5 kW to produce three consecutive locations (5 kW, 10 kW, and 15 kW). With these parameters, a request is sent to the server and returned with three circles representing catchment areas with available heads and indicated capacities for underlying households targeting tier-3 energy. Green dots are the location of proposed micro-hydropower with corresponding circular catchment areas of a selected river stream (Figure 6).

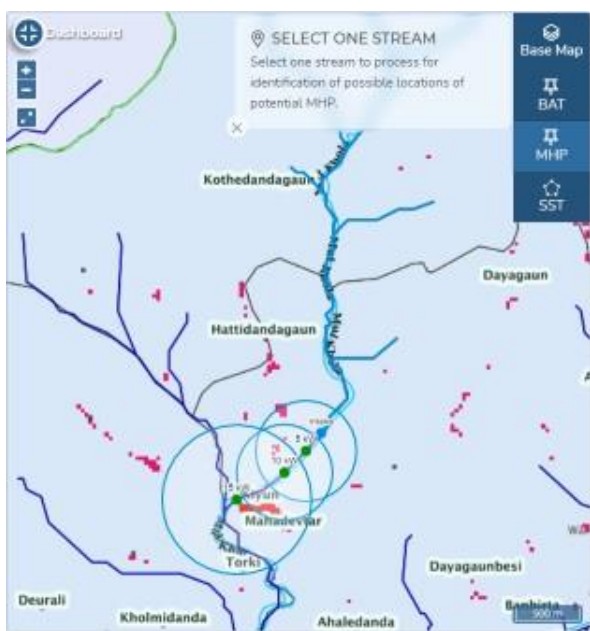

**Figure 6.** Exploring micro-hydropower (river creek line: blue; intake point: blue; circle: MHP locations: green dots; and catchment area: circular in blue).

## 5. Discussion

Energy access planning is nearly impossible in the jurisdiction of local governments with limited data availability. Data collection and analysis methods are needed to support local government decision-makers and authorities in developing evidence-based planning and for the accelerated deployment of RETs to provide energy access to both urban and rural communities. For this, the MEP toolkit has been developed using FOSS based on the latest geospatial tools and technologies and the exploitation of various geospatial datasets. Preliminarily, the toolkit uses secondary data, and later, analysis and planning are complemented by field surveys based on primary data to reflect the ground reality in the energy assessment process. Since the process has adopted a data-driven approach, the results are received with available facts and evidence. However, if the secondary data is raw or incomplete, this may be misleading. Therefore, the secondary data must be generated with field survey data towards accurate results. All the challenges due to poor road accessibility and proximity, topographic difficulties due to hilly terrains, and isolated and sparsely distributed population clusters should be considered while collecting data.

Several assumptions have been made while deriving the energy datasets and formulating methods. For example, some isolated and sparsely distributed rural settlements cannot be identified for energy assessment from census household data of 2011 [66] but can be visually inspected in OSM-based crowd-sourced building footprints were available with a question of completeness. The reasons might be (a) incomplete OSM data in rural areas and (b) old and outdated household census data. Similarly, incomplete or outdated electrical distribution lines can falsify the results for analyzing the possibility of grid extension. Moreover, for potential solar mapping, spatial resolution of 100 m has been used and solar radiation is calculated based on slope, aspect, and sun angles. One hundred meters

represents a big area that can generalize ground reality in a very undulated hilly terrain, which may not be feasible for installing SHS. Therefore, the source and quality of data being used should not be ignored while conducting energy assessments. In other words, the better the data quality the better the results. However, this toolkit can be handy for rapid energy assessments and energy planning at the municipality level. The developed methods and results show promising data that can provide local governments and authorities with the existing status-quo of energy access as well as representing a transparent starting point for the development of municipal energy access improvement plans for improving energy supply in their respective constituents.

In this paper, a FOSS-based municipal energy planning process has been demonstrated with the automatic generation of a comprehensive municipal energy plan using primary and secondary data sources. This study contributes to producing several publicly available energy datasets and opens up a new horizon for municipal energy planning at the local level. Knowing the status of energy access and gaps is critical to the energy planning process. Based on that, an energy assessment is conducted in order to help energy planners, decision-makers, and policy makers at the local level and beyond. This is an initial step for MEP in a bottom-up approach where energy assessment is performed at a village or settlement/population cluster level, and aggregated at a municipal or local level. The next step shall be operationalization of the energy planning process and scaling it to the whole region or country level.

## 6. Conclusions

Due to the mountainous topography and geographic conditions and the associated high costs of expanding the national grid, it is difficult to connect all rural/remote areas to the electricity network in Nepal. Additionally, access to reliable energy-related information and data in Nepal, particularly in the new federal structure for assessing energy planning, implementing and monitoring basic services delivery is limited. After making such data available, this data needs to be interpreted, analyzed, and processed in order to develop adequate plans and promotions programs, but the capacity of the newly created governments on local and provincial levels is low. Moreover, several studies proposed different energy assessment tools using primary and secondary data sources, e.g., decision support tool for energy planning [29], bottom-up approach for energy assessment [31], load simulation and optimization techniques [40,41,45], and cost-effective technology [42], but lacked in providing a comprehensive energy plan at the local level.

In the context of Nepal's new federal set-up, to address the challenges of the lack of capacity at the sub-national level to promote renewable energy and for better allocation of local energy resources, this paper presents a novel method with an energy assessment toolkit for geo-enabled MEP process which can support to ascertain sustainable energy access to the people. The backbone of the approach is an automatized routine-based optimization on geospatial data and techno-socio-economic input parameters. The methodology adopted in this paper constitutes optimizing electrification efforts at the local level where MEP process and GIS-based planning toolkits were introduced to determine the cost-optimal synthesis of electrification options. These tools enable the consideration of a set of energy options, including solar, micro-hydropower, and existing grid networks. Moreover, the geospatial MEP toolkit is coupled with a mobile survey app that automatically feeds collected data, e.g., existing energy infrastructure (national grid, mini-grids, off-grid technologies) data into the centralized database that helps to assess the energy situation of households, businesses, and communities. The presented approach complements the existing annual planning process of the local governments, which does not consider the geospatial characteristics of energy resources but rather is done on a more ad-hoc basis, and enables municipalities to set up targeted support programs for RETs based on sound geospatial data analysis. Additionally, it enables municipalities to collect and analyze data with minimal input from human resources as well as to promote decentralized RE by developing a comprehensive energy plan automatically. Since the tool aggregates

data, it allows municipalities, provinces, and the federal government to understand the energy-access situation, monitor development, and make informed policy decisions. The developed tools support the development of DRE and a provision of equitable energy access, thereby contributing to the SDGs with a detailed geospatial analysis of available RE options to derive an evidence-based municipal energy plan. Currently, the toolkit has been used for evaluating 30 rural municipalities of Nepal and adapted by various I/NGOs in their working areas, and will be refined as required before scaling up to the remaining municipalities in the near future. The targeted users of this toolkit are primary energy planners and decision/policy makers, and other relevant stakeholders for the energy planning decision-making process to sustainably access energy. However, the multi-year energy plan or result generated by the toolkit depends on the quality of the data input, which should not be ignored while interpreting energy assessment results and implementing the plan. Hence, it is recommended to perform quality checks of both primary and secondary data used.

**Author Contributions:** Hari Krishna Dhonju and Bikash Uprety proposed and carried out the research. Hari Krishna Dhonju proposed the MEP toolkit concept. Hari Krishna Dhonju and Bikash Uprety codesigned and developed the MEP toolkit. Hari Krishna Dhonju, Bikash Uprety, and Wen Xiao wrote the paper. All authors have read and agreed to the published version of the manuscript.

**Funding:** This research received no external funding.

**Acknowledgments:** The authors gratefully acknowledge the REEEP. The REEEP is a German—Nepali technical cooperation that is jointly implemented by Alternative Energy Promotion Center and Deutsche Gesellschaft für Internationale Zusammenarbeit (GIZ) GmbH, commissioned by Federal Ministry for Economic Corporation and Development (BMZ). REEEP strengthens the long-term promotion of RE and EE in Nepal by establishing the necessary institutional structures, processes, instruments, and coordination mechanisms at the federal and sub-national level, as well as by supporting the private sector to introduce sustainable business models and technologies while increasing private sector investment in RE and EE at the same time. Moreover, the authors also would like to thank Diwas Tamang and Suman Sanjel for their assistance in system development. This MEP process including the toolkit was developed by the Renewable Energy and Energy Efficiency Program (REEEP) to align public service delivery in the decentralized energy sector with federalized government system of Nepal.

**Conflicts of Interest:** The authors declare no conflict of interest.

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
