# Peer review of "Geo-Enabled Sustainable Municipal Energy Planning for Comprehensive Accessibility: A Case in the New Federal Context of Nepal"

_ijgi, doi:10.3390/ijgi11050304_

Round 1

Reviewer 1 Report

The draft is much improved version. However, very minor errors are still visible. Please correct those as suggested in the text.

Author Response

Comments and Suggestions for Authors

The draft is much improved version. However, very minor errors are still visible. Please correct those as suggested in the text.

Response: Thank you so much for your comments and highlighting the errors. The errors have been corrected as suggested.

Reviewer 2 Report

Dear Authors,

According to the previous review, the Abstract does not clearly capture the absolutely necessary elements: objective, method, data, data source, period, results, novelty, and usefulness of the research.

The Introduction is too long.

The period analysed is nowhere mentioned.

The Methodology should be developed by presenting concrete results to be analysed accordingly.

Conclusions are not relevant.

Author Response

Comments and Suggestions for Authors

Dear Authors,

According to the previous review, the Abstract does not clearly capture the absolutely necessary elements: objective, method, data, data source, period, results, novelty, and usefulness of the research.

Response: Many thanks for the comments. We have revised the abstract as required. The main objective of the paper is to present a geo-enabled municipal energy planning method and a comprehensive geospatial toolkit to facilitate sustainable energy access to local people in the municipalities of Nepal. It uses geospatial and socio-economic data along with household and community data collected using mobile app, rather than to presenting data. The result and novelty are a new methodology for municipal energy planning at local level with a multi-year comprehensive municipal plan generated, targeted for municipal energy planners and decision makers.

The Introduction is too long.

Response: Thanks for the comment. The introduction has been trimmed where necessary.

The period analysed is nowhere mentioned.

Response: Thanks for the comments. The methodology was developed during 2018 and 2020, now being implementing in different municipalities (rural and urban) of Nepal.

The Methodology should be developed by presenting concrete results to be analyzed accordingly.

Response: Many thanks for the suggestions. The main aim of the paper is to present the developed method which has been implemented to develop a geospatial toolkit for municipal energy planning at the local level. So, the result is a geospatial toolkit for municipal energy planning.

Conclusions are not relevant.

Response: Thanks for the comments. We revised and extended the conclusion as necessary. The conclusion is about the development of the geospatial toolkit and methods that are employed within the tool and applicability of the tool in the new federal context of Nepal.

Reviewer 3 Report

The objective of this paper is to paper to present a geo-enabled municipal energy planning method and a comprehensive toolkit to facilitate sustainable energy access to local people in a municipality in Nepal. This is an interesting topic, and the paper is fairly well written, though there are a few grammatical and typographical errors.

The introduction section paper is good. The literature review is also comprehensive, and authors have included some of the recent articles in the review.

Authors have explained well the research methodology in the materials and methods section. The methods employed are appropriate for a study of this nature.

The implementation of the proposed plan is presented clearly and analyzed appropriately in the Implementation section of the paper. Figures 2, 3, and 4 clearly shows dashboard and planning modes of municipal energy planning toolkit and the distribution of grid sample. However, Figure 5 is not clearly presented. The discussion section adequately ties together the other elements of the paper. However, authors have not compared the findings of this study with that of previous studies.

The conclusions section of the paper is very brief. Overall, the paper has expressed its case, measured against the technical language of the field and the expected knowledge of the journal's readership.

Author Response

Comments and Suggestions for Authors

The objective of this paper is to paper to present a geo-enabled municipal energy planning method and a comprehensive toolkit to facilitate sustainable energy access to local people in a municipality in Nepal. This is an interesting topic, and the paper is fairly well written, though there are a few grammatical and typographical errors.

Response: Many thanks for your comments and appreciation. We have re-proofread and grammatical and typographical errors have been corrected accordingly.

The introduction section paper is good. The literature review is also comprehensive, and authors have included some of the recent articles in the review.

Response: Thank you for the comments and feedback.

Authors have explained well the research methodology in the materials and methods section. The methods employed are appropriate for a study of this nature.

Response: Many thanks for comments and agreeing with the appropriateness of the study.

The implementation of the proposed plan is presented clearly and analyzed appropriately in the Implementation section of the paper. Figures 2, 3, and 4 clearly shows dashboard and planning modes of municipal energy planning toolkit and the distribution of grid sample. However, Figure 5 is not clearly presented. The discussion section adequately ties together the other elements of the paper. However, authors have not compared the findings of this study with that of previous studies.

Response: Many thanks for the comment and feedback. Explanation on Figure 5 has been discussed in Best Available Tool (BAT). Since the main aim of the paper is to present the methodology and geospatial toolkit, we have not compared the findings of the study with the previous studies.

The conclusions section of the paper is very brief. Overall, the paper has expressed its case, measured against the technical language of the field and the expected knowledge of the journal's readership.

Response: Many thanks for the comments and feedback. We revised and extended the conclusion as necessary.

Round 2

Reviewer 2 Report

Dear Authors,

The article is very slightly modified. I insist on the Conclusions. These should be developed by comparing your research results with those already existing, capturing the limitations, novelty and importance of your results.

Author Response

Comments and Suggestions for Authors

The article is very slightly modified. I insist on the Conclusions. These should be developed by comparing your research results with those already existing, capturing the limitations, novelty and importance of your results.

Response: Many thanks for the comments and your suggestions. The conclusions have been revised as bellow.

Due to the mountainous topography and geographic conditions, and the associated high costs of expanding the national grid, it is difficult to connect all rural/remote areas to the electricity network in Nepal. Additionally, access to reliable energy-related information and data in Nepal, particularly in the new federal structure for assessing energy planning, implementing and monitoring basic services delivery is limited. After making such data available, this data needs to be interpreted, analyzed, and processed in order to develop adequate plans and promotion programmes. But the capacity of the newly created governments on local and provincial levels is low. Moreover, several studies proposed different energy assessment tools using primary and secondary data sources, e.g., decision support tool for energy planning [1], bottom-up approach for energy assessment [2], load simulation and optimization techniques [3-5] and cost-effective technology [6], but lacked in providing a comprehensive energy plan at the local level.

In the context of Nepal’s new federal set-up, to address the challenges of the lack of capacity at the sub-national level to promote renewable energy and for better allocation of local energy resources, this paper presents a novel method with an energy assessment toolkit for geo-enabled MEP process which can support to ascertain sustainable energy access to the people. The backbone of the approach is an automatized routine-based optimization on geospatial data and techno-socio-economic input parameters. The methodology adopted in this paper constitutes optimizing electrification efforts at the local level where MEP process and GIS-based planning toolkit were introduced to determine the cost-optimal synthesis of electrification options. These tools enable the consideration of a set of energy options, including solar, micro hydropower, and existing grid networks. Moreover, the geospatial MEP toolkit is coupled with a mobile survey app that automatically feeds collected data, e.g., existing energy infrastructure (national grid, mini-grids, off-grid technologies) data into the centralized database that helps to assess the energy situation of households, businesses, and communities. The presented approach complements the existing annual planning process of the local governments, which does not consider the geospatial characteristics of energy resources but rather is done on a more ad-hoc basis, and enables municipalities to set up targeted support programmes for RETs based on sound geospatial data analysis. Additionally, it enables municipalities to collect and analyze data with minimal input from human resources as well as to promote decentralized RE by developing a comprehensive energy plan automatically. Since the tool aggregates data, it allows municipalities, provinces and the federal government to understand the energy- access situation, monitor development and make informed policy decisions. The developed tools will support the development of DRE and a provision of equitable energy access, thereby contributing to SDGs with a detailed geospatial analysis of available RE options to derive an evidence-based municipality energy plan. Currently, the toolkit has been used for evaluating 30 rural municipalities of Nepal and adapted by various I/NGOs in their working areas, and will be refined where as required before scaling up to the remaining municipalities in near future. The targeted users of this toolkit are primary energy planers and then decision/policy makers, and other relevant stakeholders for the energy planning decision making process to sustainably access to energy. However, since the multi-year energy plan or result generated by the toolkit depends on the quality of the data input, which should not be ignored while interpreting energy assessment results and implementing the plan. Hence, it is recommended to perform quality checks of both primary and secondary data used.

  1. Van Beeck, N., et al. A new method for local energy planning in developing countries. in World Renewable Energy Congress VI. 2000. Elsevier.
  2. Hiremath, R., S. Shikha, and N. Ravindranath, Decentralized energy planning; modeling and application—a review. Renewable and Sustainable Energy Reviews, 2007. 11(5): p. 729-752.
  3. Balderrama, J.P., et al., Incorporating high-resolution demand and techno-economic optimization to evaluate micro-grids into the Open Source Spatial Electrification Tool (OnSSET). Energy for Sustainable Development, 2020. 56: p. 98-118.
  4. Ciller, P., et al., Optimal electrification planning incorporating on-and off-grid technologies: the Reference Electrification Model (REM). Proceedings of the IEEE, 2019. 107(9): p. 1872-1905.
  5. Korkovelos, A., et al., Supporting electrification policy in fragile states: a conflict-adjusted geospatial least cost approach for Afghanistan. Sustainability, 2020. 12(3): p. 777.
  6. Kemausuor, F., et al., Electrification planning using Network Planner tool: The case of Ghana. Energy for Sustainable Development, 2014. 19: p. 92-101.

This manuscript is a resubmission of an earlier submission. The following is a list of the peer review reports and author responses from that submission.

Round 1

Reviewer 1 Report

Specific comments:

Line 96: Units are inconsistent ( kw/m2  and kwh/m2)

Line 285: Equation 1: RC is not defined

Author Response

Comments and Suggestions for Authors

Specific comments:

Line 96: Units are inconsistent (kw/m2 and kwh/m2)

Line 285: Equation 1: RC is not defined

Response: Many thanks for the reviewer’s comments. We have made the following changes accordingly.

The units are both corrected to kWh/m2/day.

For Equation 1, RC has been changed to “Replacement Cost”.

Reviewer 2 Report

The article has priginal research but the grammatical error needs to be corrected. Attached some directives as sample but rest of the article should be checked thorougly.

Author Response

Comments and Suggestions for Authors

The article has priginal research but the grammatical error needs to be corrected. Attached some directives as sample but rest of the article should be checked thorougly.

Response: Many thanks for the reviewer’s comments and detailed corrections. We have corrected the errors as suggested and proofread the full paper thoroughly.

Reviewer 3 Report

Dear Authors,

I read your work entitled Geo-enabled sustainable municipal energy planning for comprehensive accessibility: a case in new federal context of Nepal and I have some suggestions:

  • In the Abstract it is necessary to mention clearly where the research is carried out, the period, the results and the importance / utility of the research.
  • You specify too many keywords. If we analyze them with a simple search in the text, the words are not the axis of the paper. It is necessary to comply with the requirements of a scientific paper even here.
  • Please suggest the source of the statement in the first sentence of the Introduction. In agrarian society, did people cook with access to energy?
  • From the Abstract result that the objective of the paper is: ,,In this context, this paper examines geo-enabled municipal energy planning processes to ensure sustainable energy access to the people”. From the Introduction results that the objective of the paper is: ,, This paper aims to present the municipal energy planning (MEP) process and proposes a geo-enabled MEP toolkit in the new federal context of Nepal”. There is a difference between these goals.
  • Introduction is too long. In addition, the skeleton of the research proposal is not clear.
  • Literature Review must be developed and written accordingly. The research hypotheses and the motivation of the research are missing. These must be specified starting from the shortcomings noted in the existing literature.
  • The information on the data source is not explicit enough.
  • I don't think you explained all the acronyms used in the text.
  • You use, throughout the text, enumerations that you mark with a, b, c etc or with numbers. I recommend that you avoid this way of listing.
  • Sub-item 3.2.1 Data-Driven Approach has no place in the methodology.
  • The methodology does not apply.
  • Discussions do not apply.
  • Conclusions do not apply.
  • Most bibliographic sources are relatively old.

Researched topic is actual, but the paper has many gaps compared to the current requirements imposed by scientific research. A niche analysis does not arouse interest in the field of international research and the argumentation does not support the work at all.

Author Response

Comments and Suggestions for Authors

Dear Authors,

I read your work entitled Geo-enabled sustainable municipal energy planning for comprehensive accessibility: a case in new federal context of Nepal and I have some suggestions:

Response: Many thanks for the reviewer’s suggestions. We have carefully revised the manuscript accordingly.

  • In the Abstract it is necessary to mention clearly where the research is carried out, the period, the results and the importance / utility of the research.

Response: Thanks for the comment. The main focus of this paper is to present methodology developed for conducting municipal energy planning in Nepal. So, there is no specific period. The importance/utility of the methodology developed will assist municipality representatives and authority to prepare their energy plan, promote and ensure access to renewable energy.

We have made the changes to the Abstract as suggested.

  • You specify too many keywords. If we analyze them with a simple search in the text, the words are not the axis of the paper. It is necessary to comply with the requirements of a scientific paper even here.

Response: Thanks for the suggestion. Keywords are revised to Geospatial, Renewable Energy, Energy Accessibility, Best Available Technology, and Municipal Energy Planning as per the paper template which suggests to list 3 to 10 keywords specific to and reasonably common within the subject disciplines.

  • Please suggest the source of the statement in the first sentence of the Introduction. In agrarian society, did people cook with access to energy?

Response: Thanks for the question. Considering the relativity of the paper, the first sentence “From the very beginning of the agrarian society, energy is becoming a basic need of human beings for cooking foods.” of the Introduction has been removed.

  • From the Abstract result that the objective of the paper is: In this context, this paper examines geo-enabled municipal energy planning processes to ensure sustainable energy access to the people”. From the Introduction results that the objective of the paper is: ,, This paper aims to present the municipal energy planning (MEP) process and proposes a geo-enabled MEP toolkit in the new federal context of Nepal”. There is a difference between these goals.

Response: Thanks for pointing this out. The main aim of the paper is to examine geo-enabled municipal energy planning processes to ensure sustainable energy access to the people. This aim has been split into two sub-objectives: (a) to present the municipal energy planning (MEP) process and (b) proposes a geo-enabled MEP toolkit based on objective one to conduct MEP to achieve the main aim in the new federal context of Nepal. We have revised the Abstract and Introduction to make it clearer.

  • Literature Review must be developed and written accordingly. The research hypotheses and the motivation of the research are missing. These must be specified starting from the shortcomings noted in the existing literature.

Response: Thanks for the suggestion. We have added more related papers and strengthened the motivation after summarizing the existing work.

  • The information on the data source is not explicit enough.

Response: Thanks for the comment. Additional information on data sources has been added into the Data Source section for municipal administrative boundary, building footprints, elevation and topographic data, land cover data and population data.

  • I don't think you explained all the acronyms used in the text.

Response: Thanks for the detailed comment. We have revisited the all acronyms used and removed/edited where relevant.

  • You use, throughout the text, enumerations that you mark with a, b, c etc or with numbers. I recommend that you avoid this way of listing.

Response: Thanks for the recommendation. To be consistent, we have amended the enumeration to (a), (b) and so on as per instruction provided in the paper template.

  • Sub-item 3.2.1 Data-Driven Approach has no place in the methodology.

Response: Thanks for the comment. As the energy planning process is driven by quantitative data, we have adopted the concept data-driven approach. However, this is not the method how data are processed. To avoid further confusion, we have changed the section title to “Data Aggregation”.

  • The methodology does not apply.
  • Discussions do not apply.
  • Conclusions do not apply.

Response: Thanks for all these three comments. We have revised the method section as per suggestions above. The discussions are mostly on the data that are key to the planning process and changes have been made wherever appropriate. The conclusions have also been updated to be more concise. We think the revised sections are improved and we would appreciate further specific comments.

  • Most bibliographic sources are relatively old.

Response: Thanks for the comment. We have explored more sources and added the following papers [1-5] to the bibliography.

Researched topic is actual, but the paper has many gaps compared to the current requirements imposed by scientific research. A niche analysis does not arouse interest in the field of international research and the argumentation does not support the work at all.

Response: Thank you very much for all the comments, based on which we think the paper is well improved.

Acronyms

SDG

Sustainable Development Goals

UN

United Nations

MECS

Modern Energy Cooking Services

RE

Renewable Energy

BAT

Best Available Technology

MEP

Municipal Energy Planning

DRE

Decentralized Renewable Energy

NEA

Nepal Electricity Authority

AEPC

Alternative Electricity Authority

RETs

Renewable Energy Technologies

SRTM

Shuttle Radar Topography Mission

FOSS

Free and Open-Source Software

GeoICT

Geospatial Information and Communication Technology

MHPP

Micro-Hydropower Potential

GIS

Geographic Information System

MG

Mini Grid

SHS

Solar Home System

OSM

OpenStreetMap

LCC

Life Cycle Cost

LCOE

Levelized Cost of Energy

LCCA

Life-Cycle Cost Analysis

removed

CC

Capital Cost

OMC

Operation and Maintenance Cost

RC

Replacement Cost

FC

Fuel Cost

DR

Discount Rate

MTF

Multi-tier Framework

RGM

Report Generation Module

ODK

Open Data Kit

GDAL

Geospatial Data Abstraction Library

Additional References

  1. Mentis, D., et al., A GIS-based approach for electrification planning—A case study on Nigeria. Energy for Sustainable Development, 2015. 29: p. 142-150.
  2. Chicco, G., Data Consistency for Data-Driven Smart Energy Assessment. Frontiers in big Data, 2021. 4.
  3. Johannsen, R.M., et al., Designing Tools for Energy System Scenario Making in Municipal Energy Planning. Energies, 2021. 14(5): p. 1442.
  4. Surmonte, F., et al., A Data-driven approach to renewable energy source planning at regional level. Energy Sources, Part B: Economics, Planning, and Policy, 2021: p. 1-12.
  5. DCSD, Distribution & Consumer Services Directorate, NEA. 2020. p. 204.

Round 2

Reviewer 3 Report

Dear Authors,

The effort to apply the second form of the article is laudable, but it is obvious that your effort is minimal. Therefore, I maintain my decision, namely to reject the paper from publication.